# Discrete Element Simulation on Macro-Meso Mechanical Characteristics of Natural Gas Hydrate-Bearing Sediments under Shearing

**Meng Li** [1,2], **Hengjie Luan** [1,2,*] , **Yujing Jiang** [1,2,3] , **Sunhao Zhang** [3], **Qinglin Shan** [1,2], **Wei Liang** [4] **and Xianzhuang Ma** [1,2]

1   College of Energy and Mining Engineering, Shandong University of Science and Technology, Qingdao 266590, China
2   State Key Laboratory of Mining Disaster Prevention and Control Co-Founded by Shandong Province and the Ministry of Science and Technology, Shandong University of Science and Technology, Qingdao 266590, China
3   Graduate School of Engineering, Nagasaki University, Nagasaki 852-8521, Japan
4   State Key Laboratory for Geomechanics and Deep Underground Engineering, China University of Mining and Technology, Xuzhou 221116, China
*   Correspondence: luanjie0330@126.com; Tel.: +86-0532-86058052

**Abstract:** In order to study the macro-meso shear mechanical characteristics of natural gas hydrate-bearing sediments, the direct shear simulations of natural gas hydrate-bearing sediment specimens with different saturations under different normal stress boundary conditions were carried out using the discrete element simulation program of particle flow, and the macro-meso shear mechanical characteristics of the specimens and their evolution laws were obtained, and their shear damage mechanisms were revealed. The results show that the peak intensity of natural gas hydrate-bearing sediments increases with the increase in normal stress and hydrate saturation. Hydrate particles and sand particles jointly participate in the formation and evolution of the force chain, and sand particles account for the majority of the force chain particles and take the main shear resistance role. The number of cracks produced by shear increases with hydrate saturation and normal stress. The average porosity in the shear zone shows an evolutionary pattern of decreasing and then increasing during the shear process.

**Keywords:** natural gas hydrate-bearing sediment; direct shear test; shear mechanical characteristics; discrete element simulation; macro-meso

## 1. Introduction

Gas hydrate is a kind of by methane and water molecules under the environment of low-temperature and high pressure generated by the crystalline solid complex cage; it is widely distributed in deep-sea sediments or terrestrial permafrost regions due to its high energy density, large reserves, and pollution-free characteristics and is considered to be one of the most perspective new clean alternative energy sources in the future [1–3]. At present, the United States, Russia, Japan, India, China, and other countries have carried out gas hydrate trial mining [4,5]. However, natural gas hydrate is extremely sensitive to its occurrence conditions, and the decomposition of gas hydrate in the mining process will lead to a decrease in the shear strength of sediments and shear deformation damage [6,7] which will further induce a series of engineering disasters such as shaft wall instability, a submarine landslide, and collapse of drilling platforms [8,9]. Therefore, it is of great significance to study the shear mechanical characteristics of gas hydrate deposits to achieve the long-term stable commercial exploitation of gas hydrate.

Rock and soil mechanical tests are an effective means of studying the shear mechanical properties of hydrate deposits. Yun et al. [10] and Hyodo et al. [11] carried out a series of indoor triaxial shear tests of synthesized natural gas hydrate deposits to study the

effects of hydrate saturation, temperature, and net confining pressure on their strength and deformation characteristics. Winters et al. [12] tested the acoustic and shear properties of gas hydrate deposits drilled from the Mackenzie Delta. Santamarina et al. [13] and Yoneda et al. [14] carried out in situ triaxial shear tests of gas hydrate deposits to study the effects of different temperatures, formation pressures, and effective confining pressures on the shear mechanical characteristics of gas hydrate deposits in the South China Sea Trough in Japan. Luo et al. [15] conducted a comparative study on the mechanical properties of gas hydrate cores from the South China Sea, and gas hydrate sediment specimens were prepared in the laboratory through triaxial shear tests. Oda et al. [16] found that the effect of gas hydrate particle shapes on the sediment was mainly due to the rolling moment caused by the deviation of the normal contact force from the shape center. The results showed that the stress–strain characteristics and strength characteristics of the two were similar and confirmed that hydrate particles could enhance the cementing effect of sediment particles. The above research plays an important role in understanding the shear mechanics of natural gas hydrate deposits. However, due to the strict requirements of temperature and pressure conditions for the occurrence of natural gas hydrate, the difficulty of sampling and sample preparation, high cost, and poor repeatability meant that it was difficult to carry out a large number of experiments at the present stage.

To overcome the shortcomings of the existing laboratory test technology, scholars have carried out a series of numerical simulation studies using the particle flow code (PFC). PFC is not limited by the amount of deformation, which can easily deal with the mechanical problems of discontinuous media and can effectively simulate discontinuous phenomena such as the cracking and separation of media. Therefore, PFC can better simulate the mechanical behavior of discontinuous media such as natural gas hydrate deposits. However, at the same time, PFC also has the disadvantages of the difficult calibration of mechanical parameters in the model and a more complicated mechanical mechanism. Brugada et al. [17] simulated two preparation methods of gas hydrate deposits and found that the deviatoric stress peaks of the samples prepared by the two methods were the same. Jiang et al. [18–21] established a mesoscopic cementation model related to the mechanical properties of gas hydrate to reflect the contact mechanical response law under the cementation between the particles of gas hydrate sediments and carried out a series of discrete element simulation experiments. Jung et al. [22] also considered the cementation of the hydrates and conducted a three-dimensional discrete element simulation analysis of sediments containing two forms of hydrates. Jiang et al. [23] studied the influence of the hydrate content and loading rate on the strength, stiffness, cohesion, and internal friction angle of natural gas hydrate deposits through a biaxial shear discrete element simulation. Moreover, the evolution law of shear strength, volumetric strain, and the shear band of gas hydrate sediments under different dynamic loading conditions was studied by triaxial shear discrete element simulation [24]. He et al. [25] simulated the influence of intermediate principal stress on the mechanical behavior of natural gas hydrate deposits and demonstrated the influence of the contact rose diagram, coordination number, and damage parameters on the macroscopic mechanical behavior of natural gas hydrate deposits. Zhao et al. [26] analyzed the influence of different factors on the initial stress and damage stress of natural gas hydrate deposits and obtained the damage law of natural gas hydrate deposits. The above studies have initially revealed the macro and micro shear characteristics of gas hydrate sediments under triaxial shear conditions.

In conclusion, the current studies on the shear mechanical characteristics of hydrate deposits are mostly based on triaxial shear tests. However, the direct shear test is not only convenient and time-saving but can also quickly obtain the mechanical properties of hydrate deposits and is more suitable for describing the deformation and failure characteristics of sediments under large deformation conditions [27–29]. However, there are few studies in this area. Therefore, the direct shear numerical simulation of natural gas hydrate sediments under different saturation and different normal stress conditions was carried out by PFC in this paper, and the macro and micro shear mechanical properties and the evolution

rules of the specimens were studied, and the shear failure mechanism was revealed. The research results have certain guiding significance for the accurate understanding of the shear mechanical characteristics of gas hydrate deposits and the prevention and control of seabed geological disasters.

## 2. Discrete Element Simulation Method for Particle Flow

### 2.1. Particle Contact Model

The mechanical properties of gas hydrate deposits are affected by many factors. In addition to the well-known hydrate saturation and stress conditions, the occurrence mode of the hydrate can also significantly change its mechanical properties. Figure 1 shows three common mesoscopic distribution modes of hydrate in sediments, namely, the filling mode, skeleton mode, and cementation mode [16,29]. In this paper, the most widely distributed hydrate in nature is the filled hydrate to construct the natural gas hydrate deposit model [30]. Considering the shape effect of sand and the cementation effect between the hydrate and soil particles, the rolling resistance linear contact model and the parallel bonding contact model is adopted between the sand particles and between hydrate and soil particles, respectively [31,32].

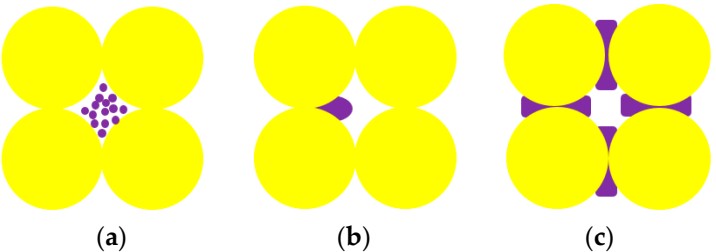

|          |          |          |
|:--------:|:--------:|:--------:|
| **(a)**  | **(b)**  | **(c)**  |

**Figure 1.** Cementation types of hydrate sediments: (**a**) padding, (**b**) skeleton, (**c**) cement. (Adapted with permission from Ref. [17]. 2010, Brugada. J., Cheng, Y.P., Soga, K).

2.1.1. Rolling Resistance Linear Contact Model

The rolling resistance linear contact model introduces the rolling resistance effect based on the linear contact model, which mainly includes the normal contact part, the tangential contact part, and the anti-rolling contact part [33], as shown in Figure 2. The linear contact model of this model includes the normal contact force $F_n$ and the tangential contact force $F_s$.

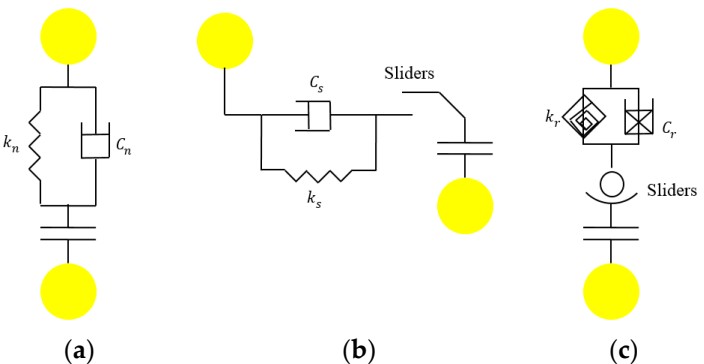

|          |          |          |
|:--------:|:--------:|:--------:|
| **(a)**  | **(b)**  | **(c)**  |

**Figure 2.** Schematic diagram of the anti-rolling linear contact model: (**a**) Normal contact part, (**b**) Tangential contact part, (**c**) Anti-rolling contact part. (Adapted with permission from Ref. [33]. 2014, Song, Y.; Yang, L.; Zhao, J).

Figure 3 shows the schematic diagram of the anti-rolling torque contact model ($k$ and $C$ represent stiffness and damping, respectively). It can be seen that in the range of $(0, \theta_r^m)$, the anti-rolling moment $M^T$ is proportional to the relative angle $\theta_r$. When the curve reaches

the turning point (point A), the anti-rolling moment $M^T$ remains unchanged at $M^*$, and the particles rotate. The specific expression is given by Equation (1) [34]:

$$\begin{cases} M^T, 0 \le \theta_r < \theta_m^r \\ M^*, \theta_r \ge \theta_m^r \end{cases} \tag{1}$$

where, $M^T$ and $M^*$ are calculated as follows [34]:

$$M^T = M^T - k_r \Delta\theta_r \tag{2}$$

$$M^* = \beta \overline{R} F_{n1} \tag{3}$$

where, $k_r$ is the rolling resistance stiffness; $\Delta\theta_r$ is the relative angle increment; $\overline{R}$ is the equivalent radius of contact. The formula for calculating $k_r$ and $\overline{R}$ is [35]:

$$k_r = k_s \overline{R}^2 \tag{4}$$

$$\frac{1}{\overline{R}} = \frac{1}{R^{(1)}} + \frac{1}{R^{(2)}} \tag{5}$$

where, $R^{(1)}$, $R^{(2)}$ is the radius of particles forming contacts.

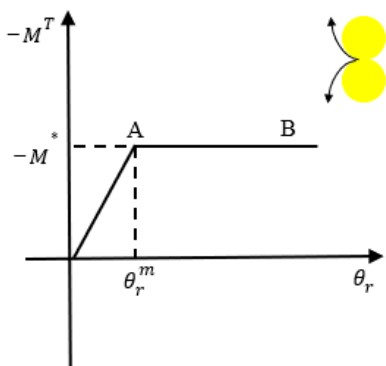

**Figure 3.** Anti-rolling torque contact model. (Adapted with permission from Ref. [36]. 2020, Wang, H; Zhou, Z.Y.; Zhou, B).

### 2.1.2. Parallel Bonding Contact Model

The parallel bonded contact model mainly consists of a cemented contact part and a linear contact part [34], as shown in Figure 4. The figure $g_s$ represents the parallel bond surface gap, and $K_n$, $K_s$ and $\overline{k}_n, \overline{k}_s$ are the stiffness between the contacts for the linear contact model and the bonded contact model, respectively; $\overline{c}$ and $\overline{\varphi}$ are the cohesive force and internal friction angle of the cemented contact model. It can be seen that the cemented contact part can be divided into the part resistant to normal tensile stress and the part resistant to shear stress. When the cemented contact part breaks, the contact model is transformed into a linear contact model, and relative sliding and friction occur between the particles.

$$\sigma^{max} = \frac{F_n}{A} + \frac{|M_b|\overline{R}}{I} \tag{6}$$

$$\tau^{max} = \frac{F_s}{A} + \beta\frac{M_t\overline{R}}{J} \tag{7}$$

where $A$, $I$, and $J$ are the cross-sectional area, the cross-sectional moment of inertia, and the polar moment of inertia in the cross-sectional section, respectively, which are calculated by the following formulas [35]: $A = \pi R^2$, $I = 0.5\pi R^4$, $J = 0.25\pi R^4$. $\sigma^{max}$ and $\tau^{max}$ are the maximum normal contact force and the maximum shear contact force, respectively. $\beta$ is the torque distribution coefficient, and this parameter defaults to 1; $\overline{R}$ is the equivalent radius

of contact. When the normal stress reaches the normal contact strength, the normal contact fails under tensile stress. When the tangential stress reaches the shear strength, the shear contact fails, and the failure condition follows the Moore-Coulomb strength criterion. The form of contact failure is shown in Figure 5.

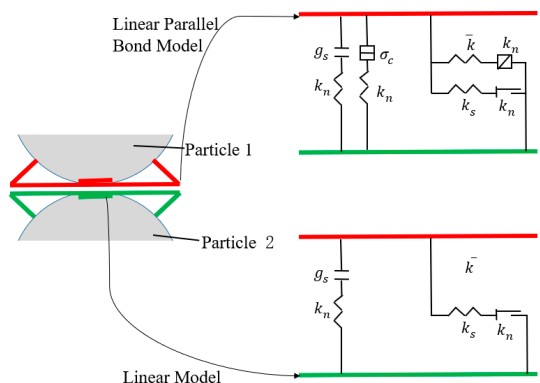

**Figure 4.** Composition diagram of parallel bond contact model.

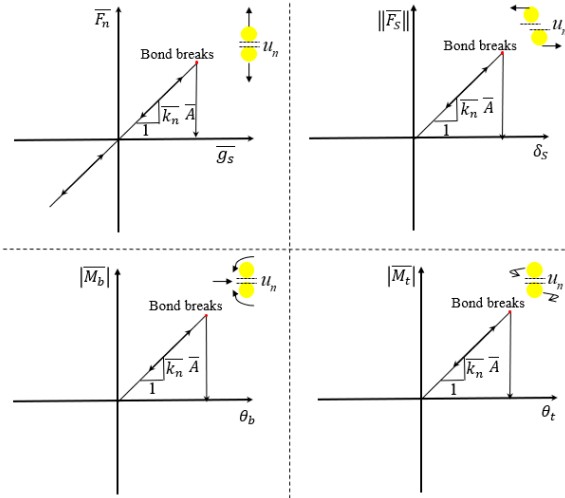

**Figure 5.** Schematic diagram of contact force displacement of parallel bonding model.

### 2.2. Determination of the Mesoscopic Parameters

The mesoscopic parameters of the particles in PFC are difficult to obtain directly, so it is crucial to establish a matching relationship between the mesoscopic parameters of the particles and the macroscopic mechanical parameters of gas hydrate deposits. In this paper, the mesoscopic parameters were selected concerning Wu Dejun's indoor triaxial shear test [37], and the specific methods are as follows:

(1) The grading that was curve used in the simulation was selected by referring to the grading curve of gas hydrate sediment particles in the Shenhu area of the South China Sea used in laboratory tests, as shown in Figure 6. Related studies have shown [37–39] that when the particle size used in the simulation is more uniform than that in the test, as long as the number of particles is sufficient, the calculation efficiency can be improved, and the mechanical response can still be obtained. Therefore, the particle size distribution type used in the simulation is similar to that in the test, but the range is narrower (0.1 μm–1000 μm).

(2) A four-side wall was established, and a 1 mm (length) × 2 mm (height) sediment specimen without a hydrate was generated according to the gradation curve, and the number of sand particles in the specimen was 5316. Related studies have shown [40–43] that when the size of a discrete element specimen is between 30 and 40 times the

average particle size, the effect of the specimen size on simulation results can be almost ignored. Considering the influence of calculation efficiency, the size of the discrete element specimen is generally smaller than that of the test specimen.

(3)  The sediment specimens were consolidated by applying consolidation pressure. After consolidation, the sand particles in the specimen were fixed, and the radius was reduced, and then the gas hydrate particles with a particle size of 0.06 mm were randomly generated between the pores of the sediment by the gradual particle size expansion method. The fixation of the sand particles is then canceled, and their diameter is expanded to the original diameter until all particles reach the equilibrium state. The self-programmed Fish function was used to generate different contact models between the different particles, and finally, the discrete element specimen of gas hydrate sediment, as shown in Figure 7, was established. It is worth noting that the number of gas hydrate particles in the specimens with different saturation is different, and the corresponding number of gas hydrate particles can be calculated by Equation (8).

$$Nhydrate = \frac{A \times n \times Smh}{\pi r_h^2} \tag{8}$$

(4)  After the model was generated, the compression test simulation of the specimen was carried out, which was consistent with the indoor test. The confining pressure was set to 0.25 MPa, the loading rate was set to 0.1 mm/min, and the loading was stopped when the axial strain of the specimen reached 25%.

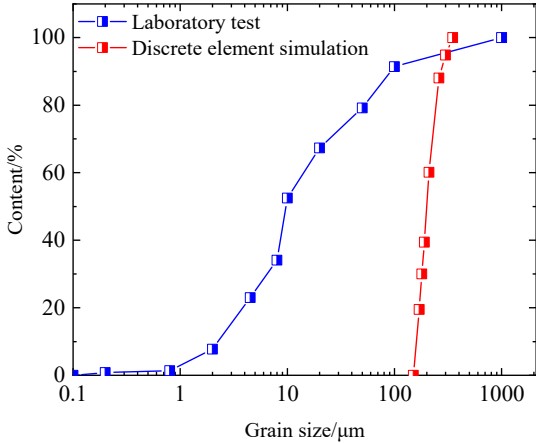

**Figure 6.** Gradation curves for gas hydrate deposits in experiments and discrete element simulations. (Adapted with permission from Ref. [37]. 2021, Wu, D.J).

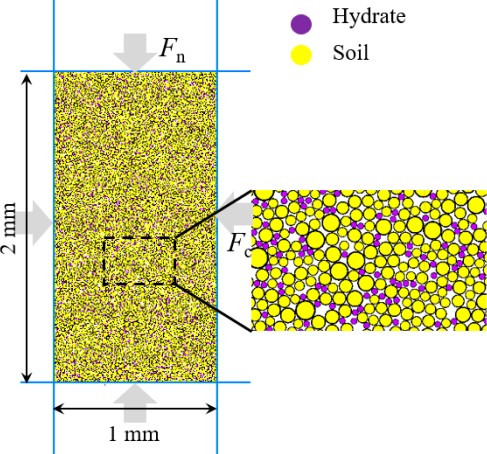

**Figure 7.** Biaxial discrete element numerical model and microstructure of gas hydrate deposits.

Based on the test results, the "trial and error method" was used to calibrate the mesoscopic parameters of the specimen. The comparison between the simulation and test results is shown in Figure 8, and the determined parameters are shown in Tables 1 and 2. The deviating-stress–axial strain curves obtained by numerical simulation and laboratory tests are in good agreement overall, and the discrete element simulation can effectively reflect the mechanical characteristics of the gas hydrate deposits. The simulation results show that the jitter in the second half of the curve is more severe, which is because, with the increase in the particle density in the compression process, the small displacement or dislocation of the particle causes stress on the monitoring wall to produce a more obvious jitter.

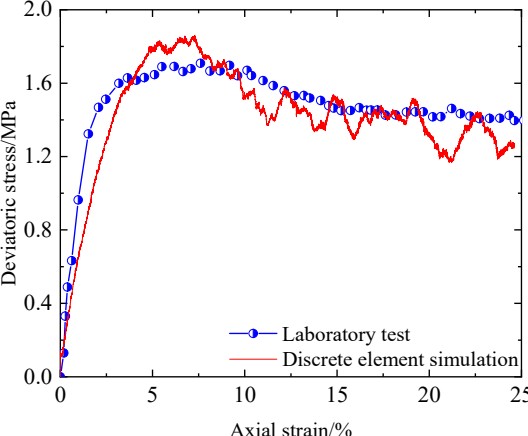

**Figure 8.** Comparison of discrete element simulation and experimental results of shear test of natural gas hydrate deposits. (Adapted with permission from Ref. [37]. 2021, Wu, D.J).

**Table 1.** Characteristic parameters of sand, hydrate particles, and walls in the model.

| Round Particles | Sand and Soil Particles | Natural Gas Hydrate Particles | Wall |
|---|---|---|---|
| Density/g·cm$^{-3}$ | 2.65 | 0.9 | |
| Normal stiffness /N·m$^{-1}$ | $2 \times 10^8$ | $2 \times 10^5$ | $1 \times 10^8$ |
| Normal tangential stiffness ratio | 1.0 | 1.0 | 1.0 |
| Grain size/mm | 0.15~0.35 | 0.06 | |
| Friction coefficient | 0.7 | 0.5 | 0.75 |

**Table 2.** Characteristic parameters of parallel bond model and rolling resistance linear contact model.

| Contact Model | Intergranular Sand and Soil | Natural gas Hydrate Sand Intergranular | Natural Gas Hydrate Interparticle | Between the Wall and the Grain |
|---|---|---|---|---|
| Normal stiffness /N·m$^{-1}$ | | $2 \times 10^5$ | $2 \times 10^5$ | $4 \times 10^8$ |
| Normal tangential stiffness ratio | | 1.0 | 1.0 | 1.0 |
| Tensile strength/MPa | | $2.7 \times 10^5$ | $2.7 \times 10^5$ | |
| Bonding strength/MPa | | $3.4 \times 10^5$ | $3.4 \times 10^5$ | |
| Friction angle/° | | 38 | 38 | |
| Rolling friction coefficient | 0.8 | | | |
| bond radii | | 0.01 | 0.01 | |

### 2.3. Direct Shear Simulation Test Scheme

Figure 9 shows the numerical model of the gas hydrate deposit used in the direct shear simulation test. The modeling method of the model is generally consistent with the numerical model in Figure 7. The difference is that the size of the specimen is 20 mm

(length) × 20 mm (height), the number of sand particles in the specimen is 10,651, and a six-wall simulation shear box is established at the boundary of the specimen. Simulation experiment for the steps: first, limit the 1 #, 3 # wall along the horizontal direction and the # 5 wall along the normal direction of the movement, and then with the help of Fish language compiled by the servo function of 2 # wall, the constant normal stress is exerted, after waiting for the calculated balance of the 4 #, 5 #, and 6 # wall in the same level of speed (1 mm/min) to exert the shear load. From the simulation results, stage IV accounts for a large proportion of the whole shear process, and the final shear displacement is set to 1.4 mm to reduce the simulation operation time. In the simulation, the hydrate saturation is 20% and 30%, and the selected normal stress is 0.9, 1.2, and 1.5 MPa, respectively.

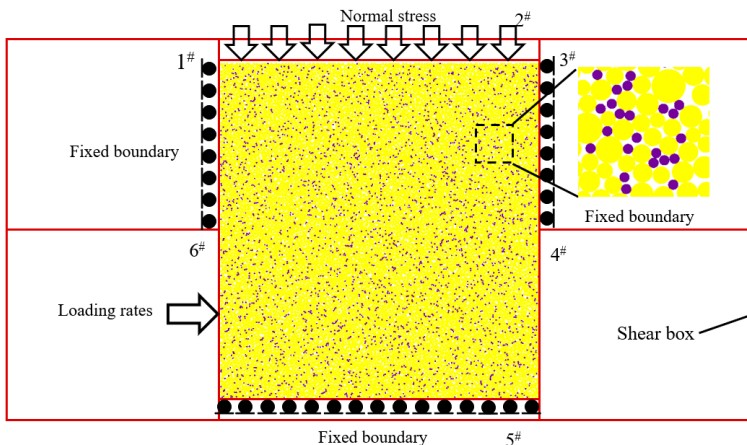

**Figure 9.** Direct shear discrete element numerical model and microstructure of gas hydrate deposits.

## 3. Macroscopic Shear Characteristics of Gas Hydrate Deposits

### 3.1. Shear Stress-Shear Displacement Curve

Figure 10 shows the shear displacement shear–stress curves of gas hydrate sediment specimens under different conditions. The variation rules of the curves are generally consistent. The simulation results under the condition of 30% saturation and 1.2 Mpa normal stress, as shown in Figure 11, are taken as an example for analysis. The shear stress–shear displacement curve can be divided into four stages. The gas hydrate sediment specimens exhibited obvious post-peak strain softening under shear force. According to the curve of normal displacement and shear displacement, except for the small shear contraction in the I stage, the dilatancy phenomenon occurred in other stages. In the third stage, the normal displacement increased significantly with the increase in shear displacement, and the dilatancy phenomenon was the most obvious, which was caused by the occurrence of more microcracks and more damage in the specimen at this stage. Compared with stage III, after entering stage IV, the microcracks and damage to the specimen were less, and the change rate of normal displacement slowed down. The reasons are detailed in Section 4.2.

### 3.2. Shear Strength Characteristic

It can be seen in Figure 10 that the hydrate saturation and normal stress have a large impact on its peak intensity. The variation pattern of the peak shear strength of gas hydrate sediments under different conditions is shown in Figure 12. The peak shear strength of the gas hydrate deposits increases with the normal stress. Taking the gas hydrate sediment specimen with 20% saturation as an example, when the normal stress increased from 0.9 MPa to 1.0 MPa, the peak strength increased from 0.66 MPa to 0.85 MPa, which was 28% higher than that of 0.9 MPa. When the normal stress continued to increase to 1.2 MPa, the peak strength of the specimen increased from 0.66 MPa to 0.85 MPa. Its peak strength increased to 1.06 MPa, which was 24% higher than that of 1.0 MPa. This is because the existence of normal stress limits the free movement of the particles in the gas hydrate

deposits of the shear process. The larger the normal stress is, the stronger the binding force on the specimen will be and the stronger the interaction force between the particles, namely the friction force between particles, which results in the greater peak strength of the gas hydrate deposits. Second, when cut to a certain extent, a specimen inevitably occurs between the soil particle sliding and rotation; sliding, under the action of the friction between the soil particles, the soil particles self-locking phenomena may occur. As shown in Figure 13, the greater the normal stress and the strength of the friction between the soil particle's, the greater the possibility of the self-locking phenomenon of soil particles and the increase in the shear strength of the specimens.

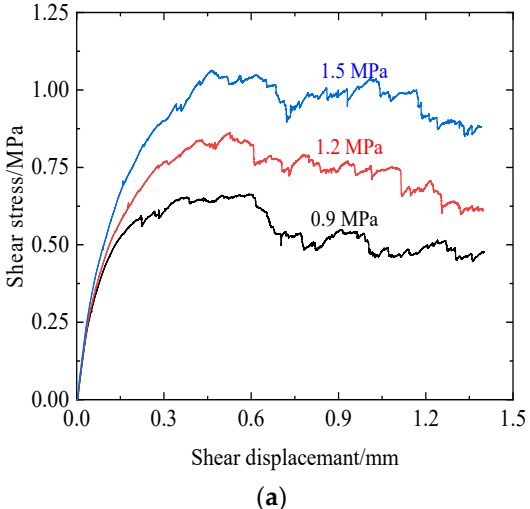 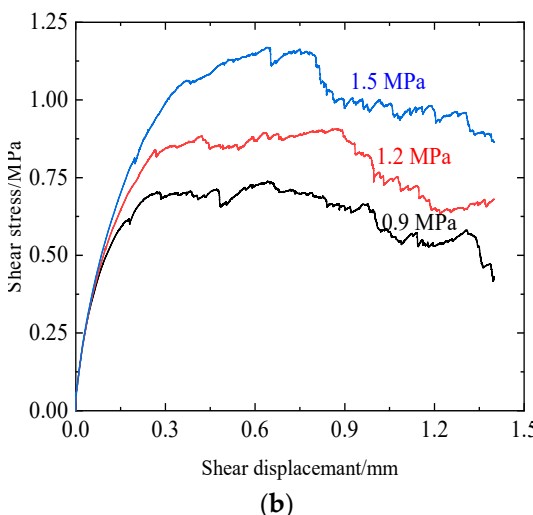

(**a**) (**b**)

**Figure 10.** Shear stress–shear displacement curve of simulated direct shear tests on nature gas hydrate deposition under different normal stresses: (**a**) $S_h$ = 20%, (**b**) $S_h$ = 30%.

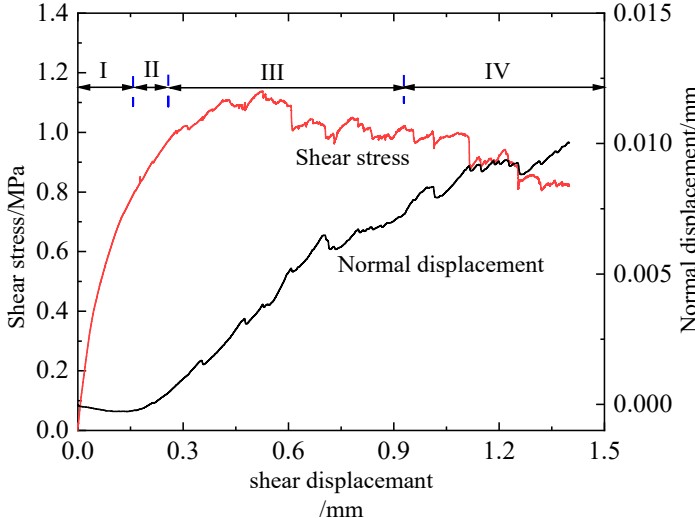

**Figure 11.** The relationship between the peak strength of nature gas hydrate deposition, $E_{50}$, and the loading amplitude.

With the increase in saturation, the cementation between particles in the gas hydrate sediment is enhanced, and its peak strength also increases slightly. Taking the gas hydrate sediment specimen under the normal stress of 1.2 MPa as an example, when the hydrate saturation increases from 20% to 30%, its peak strength increases from 1.06 MPa to 1.16 MPa. This is an increase of 9% compared to 20% saturation. However, the filled NGH deposits have obvious friction material characteristics. Although the presence of NGH can also bear

the stress of the sediment skeleton, the enhancement of the shear strength of NGH deposits is not obvious [42].

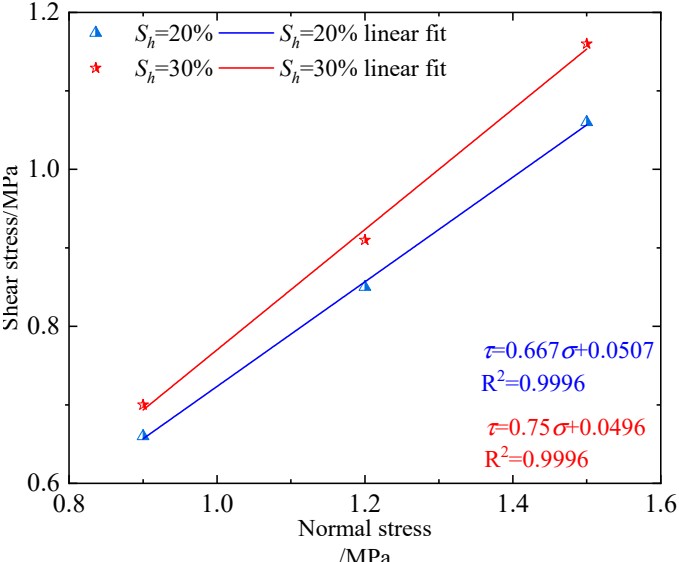

**Figure 12.** Mechanical parameters of nature gas hydrate sediments under different normal stresses.

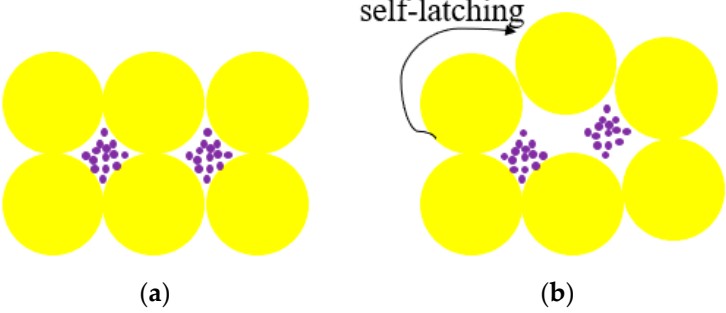

**Figure 13.** The shear mechanism between soil particles in natural gas hydrate-bearing sediments: (**a**) Before the shear, (**b**) After the shear.

From Figure 12, the strength parameters cohesion © and internal friction angle (φ) of the gas hydrate sediment specimens under different saturation conditions can also be obtained. When the saturation of the gas hydrate is 20%, the c of the specimen is 0.0507, φ is 33.7°, and when the saturation is 30%, the c of the specimen is 0.0496.9°, indicating a decrease by 3% and an increase by 9%, respectively, compared with the saturation of 20%, which indicates that the cohesion of the gas hydrate deposits under different saturations are the same. This indicates that the cohesion of the filled hydrate is independent of its saturation, but the internal friction angle increases slightly with the increase in saturation. This indicates that the shear strength of natural gas hydrate deposits under the influence of the angle of internal friction and the normal stress is larger and, compared to the size of the cohesive force of shear stress, is much smaller; the fact that this has a lesser effect on the shear strength of natural gas hydrate deposits relates to Zhou et al. [42], as the results are the same and it further reflects the filled gas hydrate sediment friction properties of the specimens.

### 3.3. Shear Displacement Field and Shear Band

To explore the gas hydrate deposits in the shear displacement of particles in the process of evolution, with saturation at 30%, under the condition of normal stress at 1.2 MPa, the gas hydrate sediment specimen, as an example, at intervals of 0.2 mm record a particle

displacement distribution cloud image (Figure 14), analysis of different shear displacements when the distribution of particle displacement occurs. We can see that:

① The upper gas hydrate sediment specimen is fixed, and the overall displacement is almost zero, only near the particles of the shear displacement; the lower gas hydrate sediment specimens under the action of shear force, on the whole, have large displacement, but in the shear plane accessories and cementation force under the action of friction between the particles, the particles near the shear plane displacement are relatively small. In turn, a large displacement gradient and local deformation appear at the junction of upper and lower shear boxes, and obvious shear bands are formed, which is a macroscopic manifestation of the localized deformation characteristics of gas hydrate deposits. With the increase in the horizontal displacement of the specimen, the shear band is also more obvious.

② At the end of the shear, the displacement difference of the gas hydrate sediment samples is mainly concentrated near the shear zone. This is because at the end of the shear plane, the interparticle cementation is broken, and the rolling and sliding occur between particles causing the friction to be reduced, which does not hinder the bottom shear moving of the specimens, the damaged part is no longer one of heritability and displacement, so in addition to the shear plane being damaged, the rest of the sediment specimen displacement distribution becomes more uniform.

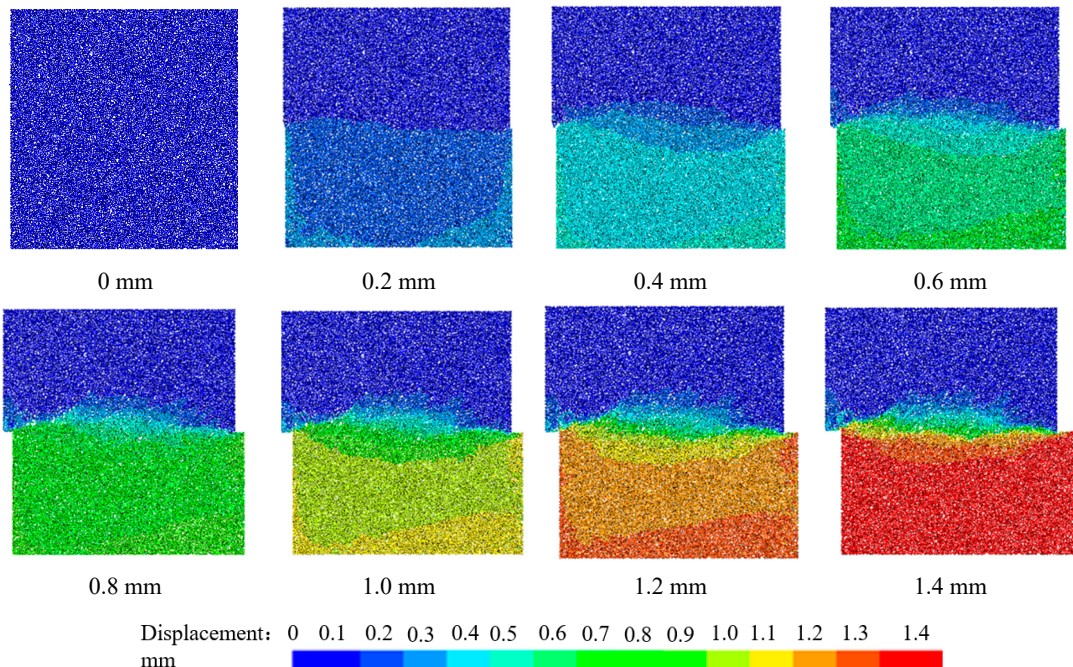

**Figure 14.** In the direct shear test, the particle displacement nephogram of natural gas hydrate sediments with normal stress of 1.2 MPa and saturation of 30%.

Figure 15 shows the shear bands of gas hydrate sediments under different saturations and different normal stress conditions when the shear displacement is 1.4 mm, and the displacement gradient varies from 0.2 mm to 1.3 mm. The shear zone of the gas hydrate deposit is greatly affected by the normal stress, and the range of the shear zone increases with the increase in the normal stress, while the range of the shear zone changes little with the increase in saturation. This is because the natural gas hydrate sediments of the shear zone from the mesoscopic level can be described as an active area of particle movement; the greater the shear strength of the natural gas hydrate sediments to resist deformation, the gas hydrate deposits within the interaction force between the particles are bigger, and the interaction force is mainly for the friction between the particles. This leads to an increase in

the area of active particle movement inside the gas hydrate deposit, and the range of the shear band increases accordingly.

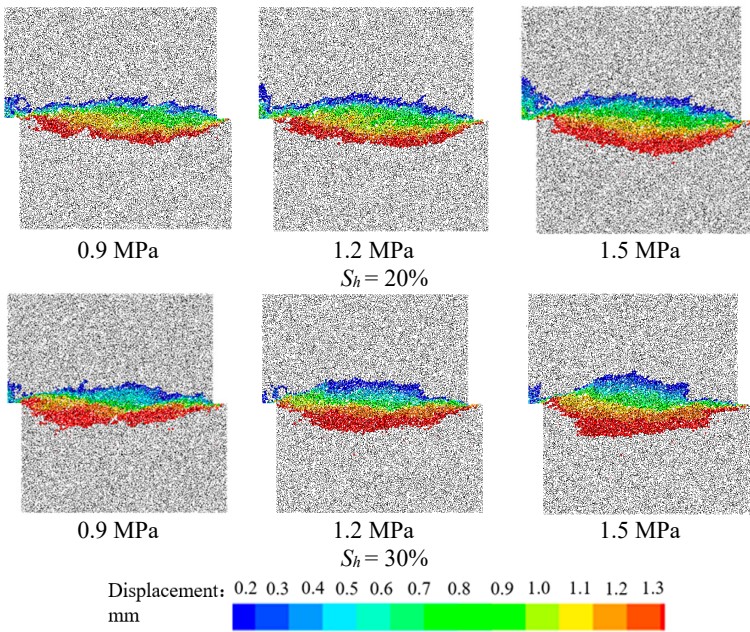

**Figure 15.** Shear zone range of natural gas hydrate deposits under different conditions with shear displacement of 1.4 mm.

## 4. Macroscopic Shear Characteristics of Gas Hydrate Deposits

### 4.1. Particle Contact State Characteristics

### 4.1.1. Contact Force Chain Network

The contact force chain network can reflect the transfer and evolution process of the contact force between the particles in the sediment specimen during shear. Taking the gas hydrate sediment specimen under the condition of 30% saturation and 1.2 MPa normal stress as an example, the contact force distribution under different shear displacements is shown in Figure 16. In the figure, the direction of contact represents the direction of force, the thickness of contact reflects the magnitude of the force, and different colors represent the interval of different contact force values. The force chain is gradually deflected as the shear proceeds. When the shear displacement is 0~0.4 mm (before the peak strength), the contact force is mainly from the left side of the lower specimen to the right side of the upper specimen, with a wide distribution range, but the maximum value is only about 1.5 KN, and the deflection direction of the strength chain (which transfers a large share of the force) is about 70° from the direction of the shear plane. After reaching the peak stress state (shear displacement 0.4~0.6 mm), several strong chains are concentrated in the force chain, the maximum value is about 2.75 KN, and the transfer direction of the force chain deviates more from the shear plane to form an angle of about 60°. When the shear displacement is 0.6~1.4 mm (after the specimen enters the IV stage), with the increase in the shear displacement, the strength chain is more concentrated in the middle of the specimen, the maximum value is about 3.4 KN, and the contact direction and the shear plane form an angle of approximately 45°~60°.

### 4.1.2. Force Chain Analysis

Force chains can reflect the transmission characteristics of the contact force inside gas hydrate sediment specimens, but it is difficult to quantitatively describe the structure and evolution behavior of contact force chains. Campbell et al. [44] and Van et al. [45] proposed the concept of force chain particles, and the selection of the force chain particles must meet the following three conditions [46]. (1. There is a threshold for the number of particles

composing the force chain, which is at least 3.2. The particles forming the force chain belong to the high-stress particles; that is, the absolute value of the minimum principal stress of the particles is greater than the average value of the absolute value of the minimum principal stress of all the particles in 3. The angle between the principal stress direction and the particle contact direction has a certain threshold, generally less than 45°).

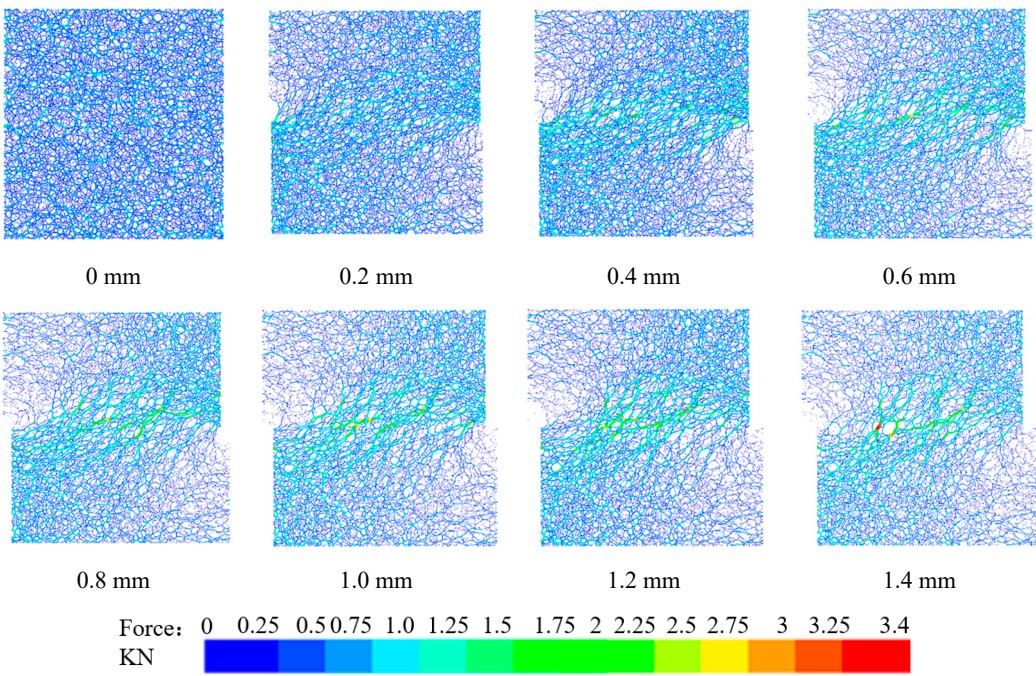

**Figure 16.** In the direct shear test, the contact force chain cloud diagram of natural gas hydrate deposits with normal stress of 1.2 MPa and saturation of 30%.

In the PFC numerical simulation software, the criterion of condition 1 is relatively easy to realize, but the criterion of conditions 2 and 3 must rely on the force analysis between the particles, as follows: under the action of normal stress, the particles inside the granular material are subjected to the interaction force of several neighboring particles, and the stress tensor of the particles can be defined [33]. In two-dimensional Cartesian coordinates, particle 1 and particle 2 are adjacent and belong to the same force chain. The stress tensor of particle 1 is given by $\sigma^1$ and its component form can be expressed as follows:

$$\sigma_{ij}^1 = \frac{1}{S^1} \sum^{n^{c,1}} \left| x_i^c - x_i^1 \right| S_i^{c,1} f_j^{c,1} = \frac{1}{S^1} \left( \sum^{n^{c,1}} R^1 S_i^{c,1} f_j^{c,n} + \sum^{n^{c,1}} R^1 S_i^{c,1} f_j^{c,t} \right); i,j = 1,2,3 \quad (9)$$

In the form: $S^1$ is the area of grain 1; $n^{c,1}$ is the number of contacts acting on grain 1; $x_i^c$ is the contact vector, $x^c$ is the component of the vector whose center of grain 1 points to the contact point; $x_i^1$ is the component of the center coordinate $x_1$ of grain 1; $S_i^{c,1}$ is the component of the unit vector $S^{c,1}$ with the geometric center of grain 1 pointing to the contact point; $f_j^{c,1}$ is the component of the contact force $f^{c,1}$; $R^1$ is the radius of grain 1.

Let the minimum principal stress of particle 1 be $\sigma_3^1$. Then, condition 2 and condition 3 can be expressed as follows, respectively

$$\left| \sigma_3^1 \right| > \frac{1}{N} \sum_{i=1}^{N} \left| \sigma_3^i \right| \quad (10)$$

$$0 < \theta = \arctan \frac{2\tau_{xy}}{\sigma_x - \sigma_y} < \frac{\pi}{4} \quad (11)$$

where *N* is the number of grains, if $\theta < 0$, then $\theta = \theta + \pi$.

Based on the above three necessary conditions, the identification process of the internal force chain in granular materials can be determined, including the screening of high-stress particles, the calculation of the direction angle, etc. It should be noted that the minimum principal stress is the maximum compressive stress on the particle because the sign of the principal stress is positive in tension.

Figure 17 shows the number of high-stress particles inside the hydrate specimen under different conditions. The number of sand particles in the force chain particles accounts for the majority: about 60~90% of the total number of force chain particles. With the increase in shear displacement, the number of force chain particles and the number of hydrate particles and sand particles in the force chain particles gradually decreased. When the shear displacement was 1.0 mm, the change in the force chain particles in the specimen gradually became stable. The higher the hydrate saturation, the greater the normal stress of the specimen, the more internal force chain particles and the more hydrate particles in the force chain particles, while the smaller the change in the sand particles of the force chain particles. The above results further illustrate the mechanical properties of filled gas hydrate sediments mainly by natural gas hydrate sediment internal friction between the soil particles, the cementation between the hydrate particle characteristics under the condition of normal stress is caused by different saturation, hydrate sediments, and the mechanical properties of different important factors.

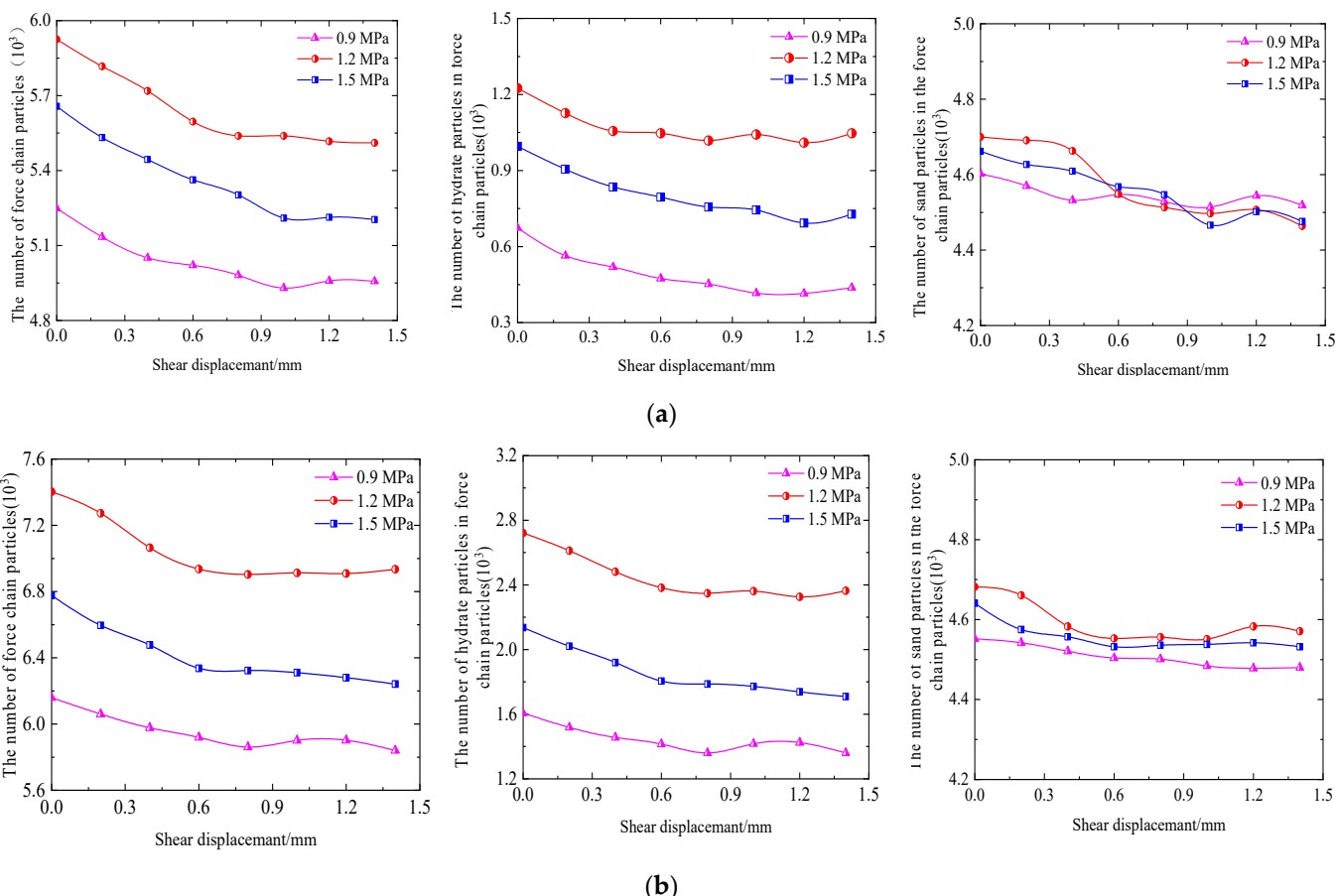

**Figure 17.** In direct shear test, the number of high stress particles of natural gas hydrate deposits under different conditions: (**a**) $S_h = 20\%$, (**b**) $S_h = 30\%$.

### 4.2. Characteristics of Crack Development and Evolution

To explore the crack development law of natural gas hydrate deposits under shear, samples with a saturation of 30% and normal stress of 1.2 MPa were taken as examples

to calculate the distribution and evolution of the cracks in the samples during shear, as shown in Figure 18. In general, the cracks are mainly distributed near the shear plane, and the number of cracks increases with the increase in shear displacement. Before the peak intensity (shear displacement) was less than 0.52 mm, the number of cracks was small and did not penetrate the whole shear plane. After the peak intensity, it can be seen that a large number of cracks are generated near the shear plane, running through the whole shear plane. From Section 4.1, we can see that: in a gas hydrate sediment specimen after shear stress is applied, the friction between the particles in the specimen begins to play the role of shear bond force near the center of a specimen, in the central particles on either side of the contact stress between the large, mainly shear effect, specimen after peak strength and the central specimen near the center of severe stress refactoring between particles. The friction and cohesion between particles decrease, the crack begins to penetrate the whole shear plane, the overall shear strength of the specimen decreases, and instability failure occurs.

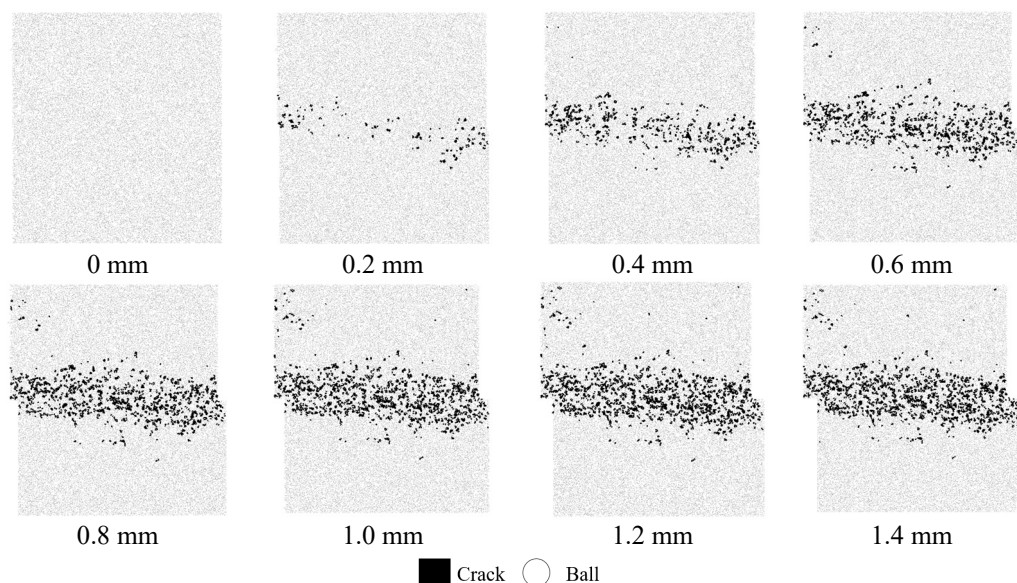

**Figure 18.** In direct shear test, crack propagation nephogram of natural gas hydrate deposits with normal stress of 1.2 MPa and saturation of 30%.

To analyze the crack evolution law of gas hydrate sediment specimens in the shear process more intuitively, the curves of shear stress, crack rate (CK_R), and cumulative crack number (CK_N) in the shear process as a function of shear displacement are given, as shown in Figure 19. As can be seen from the figure, in the first stage, the crack rate is 0, which is because the gas hydrate sediment specimen is in elastic deformation at this time, and the bond between the particles has not been destroyed. In the second stage, with the continuous application of shear stress, the cementation between the particles of the specimen was destroyed, and the specimen began to crack. However, at this time, the shear strength of the gas hydrate sediment specimen was not reached, and no large-scale failure occurred, so the crack rate was low. When entering the third stage, the bond between the particles in the gas hydrate sediment specimen was destroyed in a large number of moments, so a large number of cracks were generated, and the crack rate increased rapidly at this time. Then, as the shear continued, it entered stage IV, in which the bond between the particles near the shear plane continued to break, but compared with stage III, the crack rate began to decrease. It is worth noting that in the post-peak stage, there are still many sudden increases in the crack rate, which is due to the secondary failure of the contact between the particles after the rearrangement of the particles in the specimen.

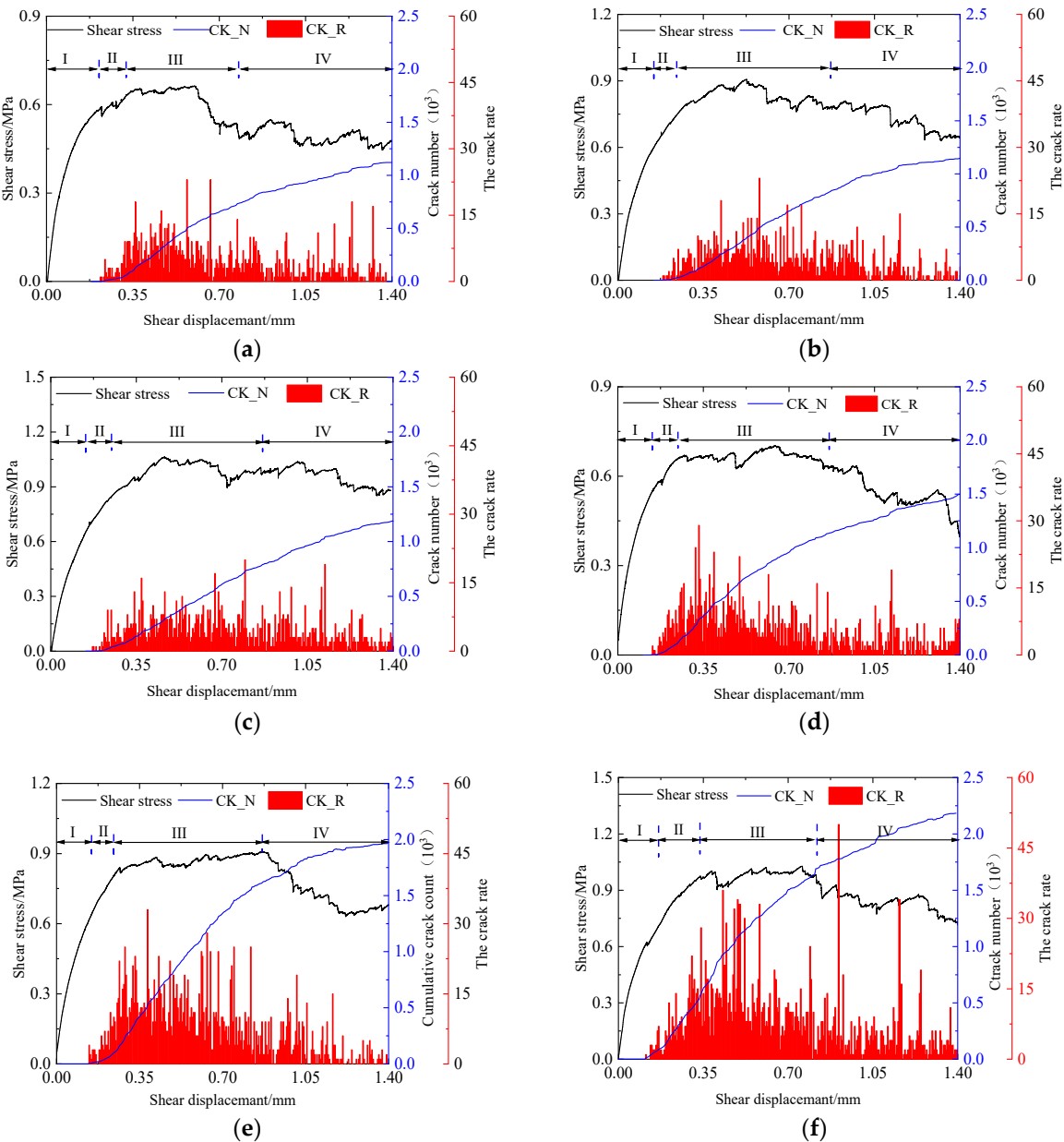

**Figure 19.** In the direct shear test, the shear stress, acoustic emission impact number of natural gas hydrate deposits under different conditions, and the cumulative acoustic emission impact number of specimens in the shear process are as follows: (**a**) $S_h$ = 20%, $\sigma_w$ = 0.9 MPa, (**b**) $S_h$ = 20%, $\sigma_w$ = 1.2 MPa, (**c**) $S_h$ = 20%, $\sigma_w$ = 1.5 MPa, (**d**) $S_h$ = 30%, $\sigma_w$ = 0.9 MPa, (**e**) $S_h$ = 30%, $\sigma_w$ = 1.2 MPa, (**f**) $S_h$ = 30%, $\sigma_w$ = 1.5 MPa.

The cumulative number of cracks in the gas hydrate deposits at the end of the shear under different conditions is shown in Figure 20. It can be seen that under the normal stress of 0.9 MPa when the saturation of the gas hydrate increases from 20% to 30%, the cumulative number of cracks increases from 1168 to 1552, which is only 32.8% higher than when the saturation is 20%. Under the normal stress of 1.2 MPa, when the saturation of the gas hydrate increases from 20% to 30%, the cumulative number of cracks increases from 1176 to 1968, which is 67.3% higher than when the saturation is 20%. However, when the saturation of gas hydrate is 20% and when the normal stress of the specimen is 0.9, 1.2, and 1.5 MPa, the crack accumulation count is 1168, 1176, 1224, respectively, and the difference between the crack accumulation counts of the three working conditions is small.

The above phenomenon indicates that when the saturation is low, or the normal stress on the specimen is small, the number of cracks in the specimen is small, and the crack rate is low (as can be seen in Figure 19), which indicates that the gas hydrate sediment shows the mechanical characteristics of the loose sand body in this state. With the increase in saturation or normal stress on the specimen, the number of cracks in the specimen is low. The crack number and crack rate of the gas hydrate deposit also increase, and the specimen shows brittleness characteristics.

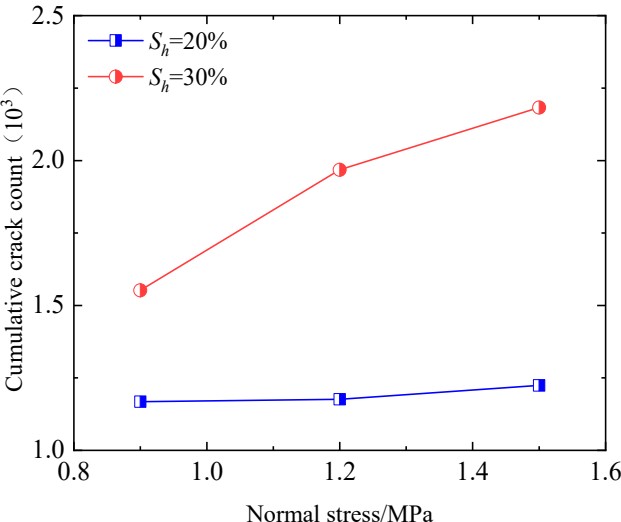

**Figure 20.** Total cracks accumulated in natural gas hydrate deposits under different conditions.

### 4.3. Evolution Law of Porosity Distribution

Porosity reflects the compactness of the particle contact in the gas hydrate sediment specimen, and its size is closely related to the specimen stress. To monitor the change in porosity in the gas hydrate sediment specimen, 100 measurement circles, as shown in Figure 21, were uniformly arranged on the specimen, and the distribution evolution law of porosity inside the specimen was obtained by monitoring, as shown in Figure 22. It can be seen that the porosity of each region shows obvious differences, and the porosity of the shear zone and the surrounding region has the largest variation, which indicates that the particle system in the region far from the shear zone has little disturbance during the shear process. In addition, after the shear test began, the porosity of the local area near the two ends of the shear plane of the gas hydrate sediment specimen began to increase and gradually expanded inward with the increase in shear displacement. Eventually, the specimen formed an obvious region of large porosity near the shear zone.

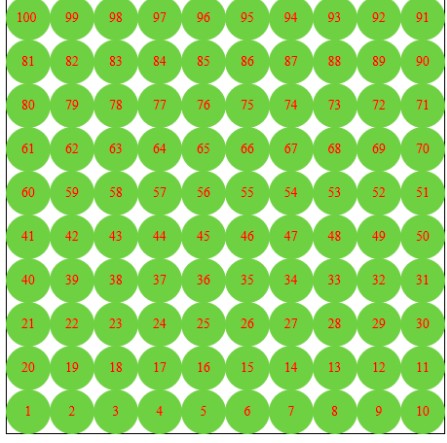

**Figure 21.** Schematic diagram of sample measurement circle layout in the direct shear test.

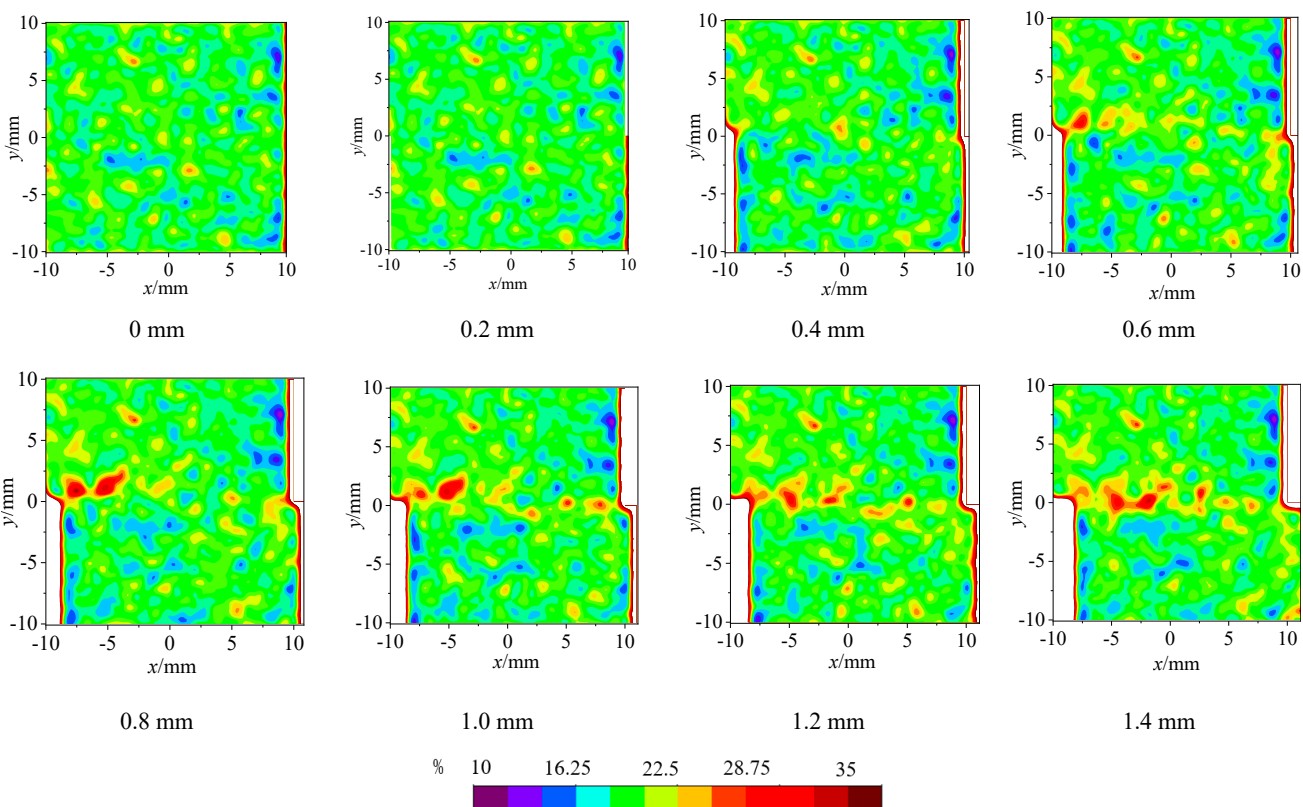

**Figure 22.** In the direct shear test, the porosity expansion nephogram of natural gas hydrate deposits with normal stress of 1.2 MPa and saturation of 30%.

To further analyze the evolution law of porosity in the shear zone, the average porosity of the measurement circles (45, 46, 55, 56) in the shear zone was counted, as shown in Figure 23. It can be seen from the figure that before the peak stress (shear displacement less than 0.54 mm), the average porosity in the shear zone of natural gas hydrate deposits showed a decreasing trend. However, after the shear zone appeared, the porosity inside the shear zone began to increase rapidly because, at this time, the intergranular cementation in the sediment specimen was largely destroyed, which further increased the porosity in the shear zone of the specimen.

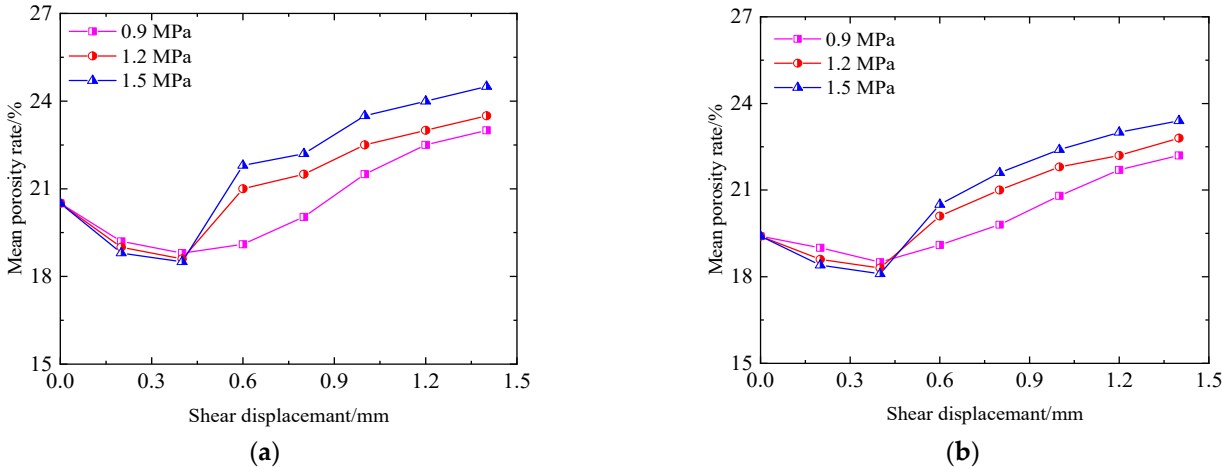

**Figure 23.** In direct shear test, the local porosity evolution curves of natural gas hydrate deposits under different conditions.: (**a**) $S_h$ = 20%, (**b**) $S_h$ = 30%.

In addition, it can be seen in Figure 23 that, taking the gas hydrate sediment specimen under the condition of 30% saturation and 1.2 MPa normal stress as an example, it can be seen that when the shear displacement is 0–0.4 mm (before the shear band is formed). The average porosity in the shear zone of the specimen changes from 19.4% when the shear displacement is 0 mm to 18.1% and when the shear displacement is 0.4 mm. The average porosity in the shear zone of the sediment specimen decreases with the increase in the normal stress. The average porosity in the shear band of the specimen is 20.1% when the shear displacement is 0.6 mm to 22.8% when the shear displacement is 1.4 mm, and the average porosity increases with the increase in the normal stress. This is because, after the peak stress and shear zone appear, the larger the normal stress to which the specimen is subjected, the larger the number of internal cracks in it, and the larger the average porosity in the shear zone of the sediment specimen.

In addition, taking the natural gas hydrate sediment specimen under the normal stress of 1.2 MPa as an example, the specimen with 20% saturation of the natural gas hydrate has an initial porosity of 20.5%, and the specimen with 30% saturation has an initial porosity of 19.4%. It can be seen that the higher the saturation of the specimen, the lower the initial porosity. When the shear displacement is 0.4, 0.6, and 1.4 mm, the average porosity in the shear band of the specimen with 20% saturation is 18.5%, 21%, and 23.5%, respectively, which is greater than that of the specimen with 30% saturation at the same shear displacement because the higher the saturation of the specimen, the greater the number of internal cracks. As a result, the average porosity within the shear band of the sediment specimen slightly decreases with the increase in saturation.

## 5. Conclusions

In this paper, the macro and micro shear mechanical properties and evolution rules of natural gas hydrate sediment specimens under different saturations and different normal stress conditions are studied, the shear failure mechanism is revealed, and the following main conclusions are obtained:

(1) The cementation between the particles in the gas hydrate deposits increase with the increases in hydrate saturation; the peak intensity of the sediment increases with increasing hydrate saturation and normal stress.

(2) In the process of direct shear, the range of the shear zone of the gas hydrate sediment increases with the increase in normal stress, but the range changes little with the increase in saturation.

(3) With the increase in shear displacement (0~0.4 mm→0.4~0.6 mm→0.6~0.8 mm), the angle between the strong chain inside the specimen and the horizontal direction gradually decreases (70°→60°→45°), and the maximum value of the contact force gradually increases (1.5 KN→2.75 KN→3.4 KN).

(4) Hydrate particles and sand particles jointly participate in the formation and evolution of the force chain, and sand particles account for the majority of the force chain particles, about 60%~90% of the total, and bear the main shear effect.

(5) The instability process of the gas hydrate deposits is closely related to the change in friction and bonding forces between the particles in the specimen, and the contact stress between the particles in the shear zone is larger than that on both sides.

(6) In the initial shear stage, the average porosity in the shear zone of the sediment specimen decreases with the increase in the normal stress, while after the peak stress and shear zone appear, the average porosity increases with the increase in the normal stress.

(7) In this paper, the effects of free gas, water pressure, and temperature on the mechanical properties of natural gas hydrate sediments were not fully considered when building the specimens; moreover, the size of the sediment specimens was set to small due to the arithmetic limitation, and the results could not be compared with the existing direct shear test results of the natural gas hydrate; future simulation studies can start

from the above two aspects to further optimize the simulation process and improve its realism.

**Author Contributions:** Conceptualization, formal analysis, funding acquisition, writing, and editing, Y.J., methodology, formal analysis, data curation, writing and editing, M.L., conceptualization, methodology, writing—review and editing, H.L., data curation, review, and editing, Q.S. and X.M., methodology, validation, S.Z. and W.L. All authors have read and agreed to the published version of the manuscript.

**Funding:** This research was funded by Shandong Provincial Natural Science Foundation, China (No. ZR2019ZD14) and the National Natural Science Foundation of China (No. 520095077).

**Institutional Review Board Statement:** Not applicable.

**Informed Consent Statement:** Not applicable.

**Data Availability Statement:** Data associated with this research are available and can be obtained by contacting the corresponding author upon reasonable request.

**Acknowledgments:** Thanks very much for the foundation of the Shandong Provincial Natural Science Foundation, China (No. ZR2019ZD14) and the National Natural Science Foundation of China (No. 52104093).

**Conflicts of Interest:** The authors declare no conflict of interest.

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
