# Peer review of "Discrete Element Simulation on Macro-Meso Mechanical Characteristics of Natural Gas Hydrate-Bearing Sediments under Shearing"

_jmse, doi:10.3390/jmse10122010_

Round 1
Reviewer 1 Report
First of all, the manuscript should be revised by a professional language editor before publishing.
Second, Heading “2. Discrete Element simulation method for particle flow” includes many redundant formulas and graphs from other references. If it is a part of your own methodology, no references is necessary. Otherwise, you could just cite the references without rewrite them again. Also, in the case of using commercial software, please just mention the detail of your sets and the chosen options. Additionally, pros and cons of the software could be mentioned. In general, the methodology is superficial and subpar. For instant, input parameters of the DEM simulation should be summarized in a table. The measuring systems and reproducibility of the study must clearly be proved with details for other interested researchers. Detail of the software is another necessary item to mention and pros and cons of the software.
Third, the number of cited references seems adequate, although they should be more internationally instead of more nationally. Some references including shape and size distribution should be cited, for example:
+ Structural and micromechanical properties of ternary granular packings: Effect of particle size ratio and number fraction of particle size classes, Materials, 13(2), 339, (2020).
Forth, there is a gas phase in this work, so why not CFD–DEM used?
Fifth, Line 375-6, how did you count the number of sand particles in the force chain particles? And why the number 60%~90%) is not fixed?
Sixth, in conclusions, item 3 is vague. A detail is needed. Additionally, conclusions must be improved by including recommendations, future work and pros and cons of the methodology.
Author Response
Comment 1: First of all, the manuscript should be revised by a professional language editor before publishing.
Reply: Thank you for this comment. We have followed this suggestion and polished the language of the paper. More details are available in the revised paper. Thank you again for this valuable suggestion.
Comment 2: Second, Heading “2. Discrete Element simulation method for particle flow” includes many redundant formulas and graphs from other references. If it is a part of your own methodology, no references is necessary. Otherwise, you could just cite the references without rewrite them again. Also, in the case of using commercial software, please just mention the detail of your sets and the chosen options. Additionally, pros and cons of the software could be mentioned. In general, the methodology is superficial and subpar. For instant, input parameters of the DEM simulation should be summarized in a table. The measuring systems and reproducibility of the study must clearly be proved with details for other interested researchers. Detail of the software is another necessary item to mention and pros and cons of the software.
Reply: Thank you for this comment. We understand the importance of removing redundant narratives, formulas and diagrams from the paper. We followed that advice and deleted some of the formulas. However, to ensure the integrity of the paper, we have retained some necessary formulas and figures. And we believe that the formulas and diagrams in Heading “2. Discrete Element simulation method for particle flow” from other literature can help the reader understand the mechanics of the contact model chosen for this paper, which has been described in detail in many papers in the relevant field (such as Zhao, Jiang, Zhou). We followed the remaining recommendations and supplemented the parameters of the DEM simulations, which should be summarized in Tables 1 and 2, and added the advantages and disadvantages of the Particle Flow Code (PFC) software. Here are modifications in the manuscript. More details of the added content are available in the revised paper. Thank you again for this valuable suggestion.
- Zhao, M., Liu, H., Ma, Q. Discrete element simulation analysis of damage and failure of hydrate-bearing sediments. Journal of Natural Gas Science and Engineering. 2022,102.
- Jiang, M., Peng, D., Ooi, J.Y. DEM investigation of mechanical behavior and strain localization of methane hydrate bearing sediments with different temperatures and water pressures. Engineering Geology. 2017,223:92-109.
- Zhou, S.C., Huan, X.L., Chen, Y.Q., Zhou, B., Xue, S.F., Gong, B. DEM simulation on undrained shear characteristics of natural gas hydrate bearing sediments. Acta Petrolei Sinica. 2021,42(01):78-83
PFC is not limited by the amount of deformation, which can easily deal with the mechanical problems of discontinuous media and can effectively simulate the discontinuous phenomena such as cracking and separation of media. Therefore, PFC can better simulate the mechanical behavior of discontinuous media such as natural gas hydrate deposits. But at the same time, PFC also has the disadvantages of difficult calibration of mechanical parameters of the model and more complicated mechanical mechanism.
|
Round particles |
Sand and soil particles |
Natural gas hydrate particles |
Wall |
|
Density/g·cm-3 |
2.65 |
0.9 |
|
|
Normal stiffness /N·m-1 |
2e8 |
2e5 |
1e8 |
|
Normal tangential stiffness ratio |
1.0 |
1.0 |
1.0 |
|
Grain size/mm |
0.15~0.35 |
0.06 |
|
|
Friction coefficient |
0.7 |
0.5 |
0.75 |
Table 1. Particle fine view parameters in the numerical model.
Table 2. Particle fine view parameters in the numerical model.
|
Contact model |
Intergranular sand and soil |
Natural gas hydrate sand intergranular |
Natural gas hydrate interparticle |
Between the wall and the grain |
|
Normal stiffness /N·m-1 |
|
2e5 |
2e5 |
4e8 |
|
Normal tangential stiffness ratio |
|
1.0 |
1.0 |
1.0 |
|
Tensile strength/MPa |
|
2.7e5 |
2.7e5 |
|
|
Bonding strength/MPa |
|
3.4e5 |
3.4e5 |
|
|
Friction angle/° |
|
38 |
38 |
|
|
Rolling friction coefficient |
0.8 |
|
|
|
|
bond radii |
|
0.01 |
0.01 |
|
Comment 3: Third, the number of cited references seems adequate, although they should be more internationally instead of more nationally. Some references including shape and size distribution should be cited, for example:
+ Structural and micromechanical properties of ternary granular packings: Effect of particle size ratio and number fraction of particle size classes, Materials, 13(2), 339, (2020).
Reply: Thank you for this comment. We have followed this suggestion and added some international references to the manuscript. Thank you again for this valuable suggestion. The additions are as follows:
- Winters, W.J., Waite, W.F., Mason, D.H., Gilbert, L.Y., Pecher, I.A. Methane gas hydrate effect on sediment acoustic and strength properties. Journal of Petroleum Science and Engineering. 2007, 56(1-3): 127-135.
- Oda, M., Konishi J., Nemat-Nasser, S. Experimental micromechanical evaluation of the strength of granular materials: Effects of particle rolling. Mechanics of Materials, 1982, 1(4): 269-283
- Joanna, W., Mateusz, S., Jalal, K. Structural and micromechanical properties of ternary granular packings: Effect of particle size ratio and number fraction of particle size classes. Materials.2020,13(2):339.
Comment 4: Forth, there is a gas phase in this work, so why not CFD–DEM used?
Reply: Thank you for this comment. We understand the importance of the effect of the gas phase on the mechanical properties of natural gas hydrate deposits. However, it has been shown that accurate results can be obtained by neglecting the effect of the gas phase in PFC simulations of natural hydrate sediment mechanics, and several relevant published examples are shown below. Moreover, the effective simulation of PFC is based on the sampling method of natural gas hydrate sediments, focusing on the interaction between natural gas hydrate and sediment particles to obtain the macroscopic mechanical properties resulting from this microscopic relationship. Therefore, the effect of gas phase is not considered in this simulation. Thank you again for this valuable suggestion.
- Yang, Q.J., Zhao, C.F. Three-dimensional discrete element analysis of mechanical behavior of methane hydrate-bearing sediments. Rock and Soil Mechanics, 2014, 35(1): 255-262.
- Tang, Q., Guo, W., Chen, H., et al. A discrete element simulation considering liquid bridge force to investigate the mechanical behaviors of methane hydrate-bearing clayey silt sediments. Journal of Natural Gas Science and Engineering, 2020, 83: 103571
- Cohen, E., Klar, A. A cohesionless micromechanical model for gas hydrate-bearing sediments. Granular Matter. 2019,21:36.
Comment 5: Fifth, Line 375-6, how did you count the number of sand particles in the force chain particles? And why the number 60%~90%) is not fixed?
Reply: Thank you for this comment. For the problem of how to calculate the number of sand particles in the force chain particles, first we set up three large groups, the first group includes 'group hydrate' and 'group soild', the second group includes 'group none' and 'group lilian'. Then write the fish function to group the particles belonging to both 'group soild' and 'group lilian' in the first group as 'group 1' in the third group, and to group 'group lilian' in the second group as 'group 1' in the third group. The particles belonging to both 'group hydrate' and 'group lilian' in the first group are coded as 'group 1' in the third group, and the particles belonging to both 'group hydrate' and 'group lilian' in the second group are coded as 'group 2' in the third group. This is used to calculate the number of sand particles in the force chain particles. In addition, we calculated the proportion of sand particles in the force chain particles in the gas hydrate sediment specimens during the simulation. Although the number of sandy soil particles in the force chain particles in the specimens with different saturation and different surrounding pressure conditions remains basically constant, the number of gas hydrate particles in the force chain particles is not fixed, and the proportion of sandy soil particles in the force chain particles under different working conditions is certainly not fixed, so we can only obtain a range of 60% to 90%.Thank you again for this valuable suggestion.
Comment 6: Sixth, in conclusions, item 3 is vague. A detail is needed. Additionally, conclusions must be improved by including recommendations, future work and pros and cons of the methodology.
Reply: Thank you for this comment. We followed this comment and revised the conclusions in the manuscript. Here are modifications in the manuscript. More details of the added content are available in the revised paper. Thank you again for this valuable suggestion.
(3)With the increase of shear displacement (0~0.4 mm→0.4~0.6 mm→0.6~0.8 mm), the angle between the strong chain inside the specimen and the horizontal direction gradually decreases (70°→60°→45°), and the maximum value of contact force gradually increases (1.5KN→2.75KN→3.4KN).
(7)In this paper, the effects of free gas, water pressure and temperature on the mechanical properties of natural gas hydrate sediments were not fully considered when building the specimens; moreover, the size of the sediment specimens was set small due to the arithmetic limitation, and the results could not be compared with the existing direct shear test results of natural gas hydrate; future simulation studies can start from the above two aspects to further optimize the simulation process and improve its realism.

Reviewer 2 Report
Very good work, original, the model is well described, the results are correct, the interpretation of the results is excellent
Author Response
Comment: Very good work, original, the model is well described, the results are correct, the interpretation of the results is excellent.
Reply: Thank you for your recognition of our work.

Reviewer 3 Report
The authors do a good job of both describing the model design, experiment and the results.
There are a few minor errors that I would like to bring to their attention.
line 46: There needs to space between the reference and the next word… Yun et al. [10]and…
line 73, 271, 272 : not sure the word Angle needs to be capitalized
There is no need for the period following Figure and before the number
Author Response
Comment: The authors do a good job of both describing the model design, experiment and the results. There are a few minor errors that I would like to bring to their attention. line 46: There needs to space between the reference and the next word… Yun et al. [10]and…line 73, 271, 272 : not sure the word Angle needs to be capitalized
There is no need for the period following Figure and before the number.
Reply: Thank you for your recognition of our work. We have followed this suggestion and revised the manuscript. Here are a few examples of modifications in the manuscript. More details of the contents modified are available in the revised paper. Thank you again for this valuable suggestion.
Figure 1 shows three common mesoscopic distribution modes of hydrate in sediments, namely, filling mode, skeleton mode, and cementation mode [15,28].
Yun et al. [10] and Hyodo et al. [11] carried out a series of indoor triaxial shear tests of synthesized natural gas hydrate deposits to study the effects of hydrate saturation, temperature, and net confining pressure on their strength and deformation characteristics.
Jiang et al. [22] studied the influence of hydrate content and loading rate on the strength, stiffness, cohesion, and internal friction angle of natural gas hydrate deposits through biaxial shear discrete element simulation.

Round 2
Reviewer 1 Report
Although the manuscript was improved, there are still some issues to consider as mentioned below.
1- There are some minor editorial and technical mistakes.
2- Figure and table captions should be clear and precise that are understandable without the text.
3- The references should correctly be revised. In the current version, there are some references that the last names and first names of authors are listed in reverse order. And some of them are incomplete. For examples, please have a look at no. 43, 45 and 46.
Author Response
Response to the comments of Reviewer #1
Comment 1: There are some minor editorial and technical mistakes.
Reply: Thank you for this comment. We have followed this suggestion and corrected minor editorial and technical mistakes in the manuscript. More details are available in the revised paper. Thank you again for this valuable suggestion.
Comment 2: Figure and table captions should be clear and precise that are understandable without the text.
Reply: Thank you for this comment. We have followed this suggestion and corrected the figure and table captions. More details are available in the revised paper. Thank you again for this valuable suggestion.
Comment 3: The references should correctly be revised. In the current version, there are some references that the last names and first names of authors are listed in reverse order. And some of them are incomplete. For examples, please have a look at no. 43, 45 and 46.
Reply: Thank you for this comment. We have followed this suggestion and corrected the problem in the references. More details are available in the revised paper. Thank you again for this valuable suggestion. Some of the references to the modification are as follows:
- WiÄ…cek, J., Stasiak, M., Kafashan, J. Structural and micromechanical properties of ternary granular packings: Effect of particle size ratio and number fraction of particle size classes. Materials.2020,13(2):339.
- Van Siclen, C.D. Force structure of frictionless granular piles. Physica A: Statistical Mechanics and its Applications, 2004,333:155-167.
- Peters, J.F., Muthuswamy, M., Wibowo, J., Tordesillas, A. Characterization of force chains in granular material. Physical review. E, Statistical, nonlinear, and soft matter physics. 2005,72(4):41307.